# Polymeric Nanorepellent Systems Containing Geraniol and Icaridin Aimed at Repelling *Aedes aegypti*

**DOI:** 10.3390/ijms23158317

**Published:** 2022-07-27

**Authors:** Lucas Rannier Melo de Andrade, Mariana Guilger-Casagrande, Tais Germano-Costa, Renata de Lima

**Affiliations:** Laboratory for Evaluation of the Bioactivity and Toxicology of Nanomaterials, University of Sorocaba (UNISO), Sorocaba 18023-000, Brazil; rannier.andrade@outlook.com (L.R.M.d.A.); marianaguilguer@gmail.com (M.G.-C.); tais_germano7@hotmail.com (T.G.-C.)

**Keywords:** nanocapsules, geraniol, icaridin, *Aedes aegypti*

## Abstract

Repellents are among the leading products used against diseases transmitted by the *Aedes aegypti* mosquito. However, their indiscriminate use or high concentrations can cause severe adverse reactions, particularly in children and pregnant women. To protect them, nanotechnology is a promising tool to encapsulate active compounds against degradation, increase their effectiveness, and decrease their toxicity, as it can promote the modified release of the active compound. This study aimed to develop polymeric nanocapsules containing the repellent actives geraniol and icaridin using low concentrations of the active component, with the objective of promoting effective activity and greater safety against adverse reactions. The nanocapsules were developed by the interfacial deposition method, and the physicochemical properties of the nanocapsules were evaluated using dynamic light scattering (DLS), nanoparticle tracking analysis (NTA), zeta potential, differential scanning calorimetry (DSC), Fourier transform infrared spectroscopy (FTIR), atomic force microscopy (AFM), release kinetics assay, and mathematical modeling. Cell viability was assessed by the MTT assay and genotoxicity analysis using the comet assay. The developed nanocapsules containing geraniol and icaridin showed mean diameters of 260 nm and 314 nm, respectively, with a polydispersity index < 0.2. The nanocapsules showed encapsulation efficiency values of 73.7 ± 0.1% for icaridin and 98.7 ± 0.1% for geraniol. Morphological analysis showed spherical nanocapsules with low polydispersity. The kinetic parameters calculated using the Korsmeyer–Peppas model indicated an anomalous release profile. Cell viability and genotoxicity analyses showed that the nanocapsules did not alter cell viability or damage DNA. The results demonstrate a promising nanostructured system with good physicochemical characteristics and good stability, with repellent activity against *Aedes aegypti*.

## 1. Introduction

Dengue, Zika, and chikungunya have become re-emerging diseases in Brazil. According to the Ministry of Health, between December/2019 and November/2020 971.136 cases of dengue were registered in Brazil, representing an incidence rate of 462.1 per 100,000 inhabitants [1,2]. Data also show a significant increase in dengue cases worldwide, from 505.430 cases in 2000 to 5.2 million cases recorded in 2019. A rise in arbovirus cases also has a social impact and a substantial economic impact on countries [3]. In 2016, approximately BRL 2.3 billion (USD 416.4 million) was spent on combating the vector (including repellents), indirect costs, and medical costs, of which approximately BRL 347 million (USD 68.8 million) was spent on the Brazilian public health system alone. *Aedes aegypti*, the primary vector, is a mosquito that belongs to the *Aedes* genus and is responsible for transmitting the dengue virus and other arboviruses, such as chikungunya, Zika, and yellow fever. Among the measures to prevent the spread of this disease is the use of repellents, which can significantly reduce the risk of bites and, consequently, the contraction of diseases transmitted by *Aedes* [2,4].

Several substances have been used in repellent products. Studies have shown that DEET (*N,N*-Diethyl-3-methylbenzamide), which was developed in 1946, is more effective than other substances on the market, but few studies have examined its safety and toxicity. Therefore, the Brazilian Health Surveillance Agency (ANVISA) restricts its use in children under two years of age [5,6,7]. The indiscriminate use of repellents containing DEET can lead to skin irritation, seizures, respiratory and gastrointestinal disorders and heart, neurological, and eye problems. Another factor linked to this substance is its deposition in the environment; studies have shown considerable concentrations of this chemical compound in environmental compartments (soil, water and air) [8,9].

Another substance used in repellent products is icaridin or picaridin, which was first developed in the 1990s. It is a colorless substance derived from pepper and is recommended by the World Health Organization (WHO) for use as a repellent and for children over two years of age. Although it has good effectiveness as a repellent, it can be toxic, and its incorrect use can increase the risk of adverse reactions, particularly in children and pregnant women [10,11,12]. Currently, the market offers repellents based on icaridin at concentrations of 10–25%, limiting its application in pregnant women and babies under six months of age. Some products also restrict their use in children under two years of age [11,13].

In the search for new substances with repellent properties, studies have shown that geraniol has significant characteristics, attracting significant interest in the pharmaceutical industry. Geraniol is a monoterpenoid found primarily in essential oils, including lemons, citronella, and roses. Studies have revealed that geraniol has several pharmacological properties, such as antifungal, anti-inflammatory, antimicrobial, repellent, and insecticidal activities [14,15,16]. However, the use of geraniol is limited by factors such as light, oxygen, humidity, and high temperatures, leading to its degradation or the generation of toxic substances, even inhibiting its repellent activity [17]. Among several substances, geraniol and icaridin are the main actives present in repellents marketed and recommended by international health authorities for being considered cosmetically more pleasant to use than DEET. Despite the wide use, the volatility of these assets creates the need for numerous reapplications to provide an effective repellent protection [18,19].

Nanotechnology can be used to protect molecules that have physicochemical limitations, including nanoencapsulation with polymers, which can help reduce volatilization and increase effectiveness [20]. These polymeric nanocapsules are carrier systems with a diameter of less than 1000 nm, a polymeric wall, and an oily core where the desired drugs or bioactive compounds can be inserted [21,22,23,24,25].

The use of nanotechnology for the encapsulation of molecules aimed at repellent applications is a promising strategy because the nanoencapsulation process, in addition to protecting molecules from degradation (physical or chemical), is also a system that can promote the controlled release of these molecules, providing greater safety and effectiveness. Kelidaril et al. [26] developed and characterized solid-lipid nanocapsules containing essential oil from *Zataria multiflora* for repellent applications against the *Anopheles stephensi* mosquito. The developed formulations had a particle diameter of 134 ± 7 nm. It was also observed that the formulation containing the nanocapsules had a protection/repellence time three times longer than that of the non-nanoencapsulated essential oil. The study also revealed that solid-lipid nanocapsules did not present toxicity against HFFF2 cells, thus developing a promising nano-formulation with repellent activity. Gelatin nanocapsules containing essential oils from *Piper aduncum* L. and *Piper hispidinervum* C. were developed and evaluated against *Aedes aegypti* mosquitoes. The nanocapsules had a mean diameter of 100 (±2) nm, a zeta potential between (–43.5 ± 3) mV and (–37.5 ± 2) mV, and showed efficiency against the *Aedes aegypti* mosquito after 24 h of exposure [27]. Abrantes et al. [28] developed and evaluated the permeation of nanostructured lipid carriers loaded with a mixture of icaridin (synthetic) and geraniol (natural), incorporated into cellulose hydrogel, which presented an average particle size of 252 ± 5 nm in addition to a low flux of the active agents through the membrane evaluated in the permeation test, giving the system a good product with repellent action using nanotechnology.

This study aims to develop nanocapsules of polycaprolactone (PCL) containing geraniol and icaridin, providing a modified release resulting in longevity of the repellent action, activity at a lower concentration of the active component, better safety, and reduction in adverse reactions, in addition to evaluating its characteristics. After the development, the physicochemical properties and stability of the formulation will be evaluated, and the analysis of cytotoxicity and genotoxicity of polymeric nanocapsules in cell lines V79 (Chinese hamster normal lung cells) and 3T3 (Swiss albino embryonic cells) will be performed. This study is expected to provide alternative and safe formulations for use as mosquito repellents, particularly against *Aedes aegypti*.

## 2. Result and Discussion

### 2.1. Characterization of Nanocapsules

The developed formulations had a liquid and milky appearance (Figure 1A) without visible changes such as sedimentation or flocculation. The pH of the polymeric nanocapsule suspensions was analyzed over 90 days using a pH meter (OHAUS Starter3100) calibrated with buffer solutions at pH 7.0 and 4.0. During the first 90 days, the pH was between 4.3 and 5.9 (Figure 1B). The analysis of pH is a crucial parameter for evaluating the stability and compatibility of formulations. Large variations in pH can influence stability and indicate the presence of chemical reactions or bacterial growth, which can compromise the quality and effectiveness of the formulations [29,30]. In a study conducted by Souza et al. [31], where PCL nanocapsules containing herbicides were developed and evaluated, the pH values were found to be between 4.8 and 5.6 during a 30-day stability test, similar to the values obtained in this study.

Nanocapsules containing geraniol and icaridin had a hydrodynamic diameter and an initial size close to 300 nm (Figure 2), indicating favorable reproducibility in the development process. During the storage period, the formulations showed minimal variation in diameter. The formulation Nano_Control (without bioactive compounds) showed variation after 15 days of storage. However, at 90 days, there was no significant variation. The Nano_GER3% formulation showed a significant increase in diameter during 30 days of storage from that at time 0; however, no further change was observed up to 90 days.

The Nano_GER1% formulation presented an initial diameter of 326 ± 6 nm, and this value decreased with time. The sample Nano_ICA3% showed an initial diameter of 284 ± 3 nm, which increased to 298 ± 2 nm after 90 days; however, the values at 15, 21, 30, and 60 days were significantly decreased from that at day zero. This result may be related to the degradation of the polymer and release of the active material. The nanostructured systems are of a desirable size for topical application, which is essential for the retention of nanocapsules in the epidermis, where particles with diameters <100 nm have a greater possibility of reaching deeper layers of the skin [29]. While evaluating the polydispersity index (Figure 3), it was observed that until the maximum evaluation period (90 days), all the developed formulations presented PDI values lower than 0.2. According to Danaei et al. [32], PDI values below 0.2 are considered acceptable for polymeric nanocapsules, indicating the presence of a homogeneous and monodispersed population, giving the formulation good stability and avoiding the possibility of particle aggregation.

The average size of the nanocapsules and the particle concentration as a function of time (0, 8, 15, 21, 30, 60, and 90 days) for each formulation (Figure 4) were obtained using the nanoparticle tracking analysis (NTA) technique. The sample (a) Nano_Control at time 0 had a hydrodynamic diameter of 196 ± 5 nm and a concentration of 1.6 × 10^13^ ± 5.37 × 10^11^ particles/mL. For the formulations (b) Nano_GER1% and (c) Nano_GER3%, the particles were 220 ± 12 nm and 184 ± 7 nm, with concentrations of 1.39 × 10^13^ ± 6.4 × 10^11^ and 9.8 × 10^12^ ± 1.1 × 10^12^ particles/mL, respectively. The formulations containing the bioactive icaridin, (d) Nano_ICAR1% and (e) Nano_ICAR3%, had initial hydrodynamic diameters of 184 ± 5 and 205.4 ± 7 nm, 1.9 × 10^13^ ± 9.7 × 10^11^, and 1.9 × 10^13^ ± 1.4 × 10^12^ particles/mL, respectively.

Over the 90 days, the Nano_Control sample had an average diameter of 165.6 ± 4 nm and a concentration of 4.10 × 10^13^ ± 8.99 × 10^11^ particles/mL. The formulations Nano_GER1% and Nano_GER3% after 90 days had mean diameters of 178 ± 8 and 192 ± 3 nm, and concentrations of 3.2 × 10^13^ ± 3.74 × 10^11^ and 1.3 × 10^13^ ± 3.6 × 10^11^ particles/mL, respectively (Table 1). The sample Nano_ICAR1% had an average diameter of 178.1 ± 8.4 nm and a concentration of 2.6 × 10^13^ ± 8 × 10^11^ particles/mL over 90 days, while the formulation Nano_ICAR3% had an average diameter of 206 ± 9 nm and concentration of 2.8 × 10^13^ ± 3 × 10^11^ particles/mL.

The results show that the nanocapsules present lower diameters than those obtained using DLS. According to Filipe et al. [33], this difference in values can be explained by the size distribution in DLS based on mass distribution, unlike the values obtained by NTA, which are based on numerical distribution. Maruyama et al. [34] observed a slight difference in the size of nanocapsules when developing polymeric nanocapsules containing two different nanocapsules of herbicides. The difference in values obtained between these techniques may be associated with the analysis process by NTA, in which the samples undergo a greater dilution than that used in the DLS; consequently, this dilution can cause the rupture of aggregates resulting in average values of particle distribution slightly smaller than those obtained by the DLS technique [35].

### 2.2. Differential Scanning Calorimetry (DSC)

The DSC profile of the PCL polymer shows a narrow (Figure 5) endothermic peak at 59 °C, representing the polymer’s melting point and, thus, supporting its crystallinity. Because it is a thermoplastic polymer, PCL has a low melting index, starting at approximately 50–55 °C (43). The Nano_Control formulation shows an endothermic peak at approximately 48 °C, corresponding to the polymer melting temperature; a second peak with minimal intensity was detected at approximately 129 °C. The formulations containing the bioactive icaridin show endothermic peaks at 49.6 and 139 °C for the Nano_ICAR1% formulation and 46.8 and 146.1 °C for the Nano_ICAR3% sample; the first peak is related to the melting point of the PCL polymer and the second peak to the surfactant Span 60; however, the presence of a third, broader peak may be related to the bioactive icaridin. The shift in the temperature ranges of the PCL polymer at different concentrations of bioactive compounds indicates the incorporation of active compounds into the polymer matrix.

Lee et al. [36] developed nanocapsules using a PCL polymer for the encapsulation of pilocarpine and observed a characteristic peak of the polymer at ~59 °C. In their analysis of the nanostructured system, the peak at approximately 198 °C was related to the pilocarpine, and it was pointed out that the peak at 59 °C was related to the destructuring and fragmentation of the ester group of the polymer. The DSC results obtained by Zanetti et al. [37] suggest that the decrease in the enthalpy of fusion of nanostructured systems promotes the presence of a physical bond between the ester molecules and PCL matrix. Santos et al. [38] performed a colorimetric analysis of a 𝛽-Cyclodextrin complex incorporating the essential oil of *Cymbopogon winterianus,* which has geraniol as a significant component, and obtained an endothermic peak at 152 °C related to the decomposition point of the oil after the complex was formed. The temperature values obtained from the nanostructured systems conferred great thermal stability to the bioactive geraniol and icaridin, thus indicating the incorporation of these bioactive compounds into the polymer matrix.

### 2.3. Fourier Transform Infrared Spectroscopy (FTIR)

The spectra (Figure 6a) obtained from the bioactive geraniol and icaridin. The analysis showed bands at 3326 cm^−1^ corresponding to the OH group, and bands at 2967 and 2916 cm^−1^ are related to the stretching vibrations of alkane C-H bonds present in the structure of the bioactive geraniol. The spectra obtained from icaridin show a band at 3450 cm^−1^ corresponding to stretching of the OH group, the band at 2938 cm^−1^ is characteristic of alkyl groups, and the peak at 1667 cm^−1^ corresponds to the angular deformation of C-O. For icaridin, the band at 3450 cm^−1^ is due to stretching of the O-H group, the band at 2938 cm^−1^ is a characteristic band of the alkyl group, and the peak at 1667 cm^−1^ is assigned to angular deformation of carbonyl group (CO) [39].

The spectra of the developed nanoformulations are represented in Figure 6b. In the spectrum of the PCL polymer, the band at 1744 cm^−1^ represents the stretching vibration of the C=O group of amides, and the band at 1462 cm^−1^ is due to angular deformation of the CH_2_ group. In analyzing the Nano_Control formulation, the bands at 3398 cm^−1^ correspond to the O-H functional group because of moisture present in the formulations, and the bands at 2928 and 2856 cm^−1^ are respectively attributed to the asymmetric and symmetric axial deformation of C-H of the CH_2_ group. Another band observed at 1744 cm^−1^ represents the stretching vibration of the C=O group of amides, and the band at 1462 cm^−1^ is due to angular deformation of the CH_2_ group. It can be seen that the other spectra obtained from the nanoformulations containing the bioactive substances geraniol and icaridin show bands similar to those obtained from the Nano_Control formulation. This result suggests that the spectra of geraniol and icaridin overlap with the characteristic bands of PCL, indicating encapsulation of the active compounds.

### 2.4. Encapsulation Efficiency

The nanoemulsion systems Nano_GER1% and Nano_GER3% showed an encapsulation efficiency of 98.7% ± 0.01% and 98.7% ± 0.004%, respectively (Figure 7), without significant variations (*p* < 0.05) throughout assessment. The nanocapsules containing icaridin had values of 52.5% ± 3.7% for the Nano_ICAR1% formulation and 73.7% ± 0.1% for the Nano_ICAR3% sample during the stability evaluation period, with only slight variations in the values. Through this analysis, it was observed that the nanocapsules containing geraniol showed a higher encapsulation efficiency than the nanocapsules containing icaridin, which may be related to the difference in solubility between geraniol (100 mg/L 25 °C) and icaridin (8.6 g/L 20 °C). Factors such as the affinity of the drug for the polymer, volume of the hydrophobic surface, and solubility of the active ingredient in water can affect the encapsulation efficiency of nanoparticulate systems [40]. Based on the results obtained so far, only the Nano_GER3% and Nano_ICAR3% formulations were selected for further characterization tests, evaluation of cytotoxicity, genotoxicity, and release test because, among the samples developed, these presented the most satisfactory results in terms of encapsulation efficiency and physicochemical stability.

### 2.5. Atomic Force Microscopy (AFM) 

The Nano_GER3% (Figure 8) and Nano_ICAR3% (Figure 9) samples were selected for morphological analysis of the nanocapsules by AFM analysis.

Figure 8 shows the results obtained by AFM analysis for Nano_Ger3%, displaying spherical structure and a homogeneous size of approximately 198 nm. These data corroborate the PDI values determined by DLS, although there is a difference in the size of the nanocapsules compared to the diameter analyzed by DLS. This can be attributed to several factors such as aggregation or flattening of the nanocapsules when they are deposited and dried on the mica surface, giving rise to heterogeneous populations [41,42].

Figure 9 shows that the nanocapsules in the Nano_ICAR3% formulation are also spherical, homogeneously distributed, and have an average size of approximately 148 nm. The difference in the diameters of the nanocapsules as analyzed by DLS and AFM can be attributed to several factors, such as aggregation or flattening of the nanocapsules when they are deposited and dried on the mica surface, thus giving rise to heterogeneous populations [42].

In summary, the AFM results show that the nanocapsules containing icaridin and geraniol present spherical morphology, as desired. The diameters are smaller than those obtained using DLS, which can be attributed to the dilution and drying processes necessary to perform the analysis.

### 2.6. In Vitro Release

Figure 10 shows the chemical structures of the PCL polymer and the actives geraniol and icaridin. The release kinetics profile (Figure 11) shows that the Nano_ICAR3% sample released 50% of its activity in approximately 75 min, whereas the Nano_GER3% formulation released only 7% of its activity during the same time, reaching a release of only 26% after 24 h. This result is directly related to the encapsulation efficiency profile. The nanostructured system containing icaridin showed a low encapsulation efficiency, therefore, a faster release was attributed due to a higher concentration of free active (not encapsulated). Similar release profiles were found for the prepared geraniol and icaridin emulsions. The Emulsion_ICAR3% sample released 50% of its activity within approximately 47 min. This difference in the rate of release may be directly related to the interaction of the active substance with the polymer matrix, which delays the release compared to that in the emulsion.

The nature of the polymer is an important factor in nanostructured systems. Such as the polymer used in these systems is hydrophobic, such as PCL, the release of the active is controlled by surface erosion, and upon reaching hydrophobic and hydrophilic balance, the release proceeds through the hydrolysis of ester bonds in the backbone, as well as by enzymatic attack [43,44]. According to the results obtained by FTIR (Section 2.3), there was no interaction between the actives and the polymeric matrix, which means the effectiveness of the encapsulation process. The non-interaction between the polymer matrix and the encapsulated assets is important for a controlled release to occur.

The release profiles of the polymeric nanocapsules were evaluated using Higuchi, Korsmeyer–Peppas, Hixson–Crowell, and first-order kinetic models (Table 1). Regarding the systems containing active geraniol, it was observed that the nanocapsules and the emulsion presented a similar release profile. This may be associated with the formation of micelles that, when interacting with the lipophilic membrane, have an affinity for the receptor medium, leading to the passage of the repellent. This result may be related to the difference in partition coefficients of the repellents: geraniol (LogP 3.56) and icaridin (LogP 2.11 at 20 °C).

Release kinetic models are obtained through mathematical interpretations. Models for the release of drugs and active molecules are essential to predict the mechanism of release and the concentration at the active site, which is crucial for the effectiveness of nanostructured systems [45]. The Hixson–Crowell model suggests that the release of the compounds is limited by the dissolution of the particles in which there is a change in the diameter and the surface area of the nanocapsules, which can be attributed to the principle of the release of active compounds at the rate of erosion of the polymer matrix [46]. When adjusting the release kinetics for a first-order model, Nano_GER3% showed a coefficient of determination R^2^ = 0.998; that is, the release rate was dependent on the concentration of the encapsulated molecule. Similar values obtained for the release constants of the nanocapsules containing geraniol and icaridin suggest that the release of these active molecules occurs in a single step without the presence of a burst effect, which may indicate that the entire molecule is encapsulated, in correlation with the encapsulation efficiency values [47].

### 2.7. Evaluation of Cytotoxicity

The evaluation of the cytotoxic effects of the nanocapsules through the MTT assay indicated mitochondrial activity of the 3T3 and V79 cell lines after exposure to the formulations (Figure 12). For this evaluation, the formulations Nano_Control and Nano_GER3%/ICAR3% were used, resulting from a (1:1) mixture of the Nano_GER3% and Nano_ICAR3% formulations. The cytotoxicity results of Nano_Control (Figure 12a) indicated that cells had high cell viability in both cell lines when exposed to nanocapsules concentration. At the lowest concentration, equivalent to 0.00009 mg/mL, it was possible to observe cell viability above 90%; this viability decreased as the concentration increased. The 3T3 cell line showed cell viability greater than 50% up to a concentration of 0.0025 mg/mL. The results demonstrate that the empty nanocapsules did not significantly affect cell viability in cell lines 3T3 and V79. PCL, the coating polymer that forms the polymeric matrix of the developed nanocapsules, is a biocompatible and biodegradable material that is widely used to develop micro- and nanospheres for biological applications.

Queiroz et al. [48] evaluated the cytotoxicity of geraniol using the MTT method in human hepatocyte (HepG2) cells and showed a decrease in cell viability of less than 70% at a concentration of 25 µg/mL (0.025 mg/mL). This result could be related to the inhibitory effect of geraniol (50 and 200 µM) on the mevalonate pathway and phosphatidylcholine biosynthesis. The cell viability results of the 3T3 and V79 cell lines against the Nano_GER3%/ICA3% formulation after exposure for 24 h showed that the 3T3 cell line, initially at the lowest concentration, presented a viability of 66%. In contrast, there was a 50% viability at a concentration of 0.0025 mg/mL and a viability of 43% at a concentration of 0.02 mg/mL. For the V79 strain, 100% and 65% cell viability were obtained at concentrations of 0.0005 and 0.005 mg/mL, respectively. However, unlike the 3T3 strain, 50% viability was reached at 0.0075 mg/ mL (Figure 12b).

### 2.8. Evaluation of Genotoxicity

The comet assay is a test widely used to evaluate the genotoxicity of nanocapsules, and it is capable of detecting single and double DNA breaks that are identified by damaged and negatively charged low-molecular-weight DNA fragments, forming a trail similar to a comet tail; the greater the damage, the greater the intensity of the tail [49]. Genotoxicity analyses using the 3T3 cell line exposed to Nano_Control (S/Active) nanocapsules and those containing geraniol and icaridin (1:1) exposed for 1 h did not show a significant increase in DNA damage in the negative control (Figure 13). Similar results were found for the V79 cell line, which did not show significant variation in the negative control group, indicating that the nanostructured system did not cause damage to the DNA of the lines used.

The use of the comet assay in the assessment of genotoxicity is subject to variable responses owing to the use of different methods and cellular systems, which may vary in metabolic capacity and DNA repair [50]. The genotoxicity results of nanocapsules containing geraniol and icaridin against cell lines 3T3 and V79 showed that the nanostructured systems did not show significant DNA damage compared to the negative control, which is indicative of low DNA damage. However, the evaluation of nanomaterials in biological systems requires a greater variety of assays to assess their cytotoxicity and genotoxicity to assure that this technology is safe when exposed to humans and the environment.

## 3. Methods and Materials

### 3.1. Materials 

Poly-ε-caprolactone, Tween 80 (mean micellar molecular weight, 79 kDa), and sorbitan monostearate surfactant (Span60^®^) were obtained from Sigma Aldrich (St. Louis, MO, USA). Capric/caprylic acid triglyceride (Myritol 318) was obtained from LabSynth (Diadema, SP, Brazil). Geraniol and icaridin were obtained from Sigma Aldrich. Acetone p.a. was obtained from Dinâmica Química Contemporânea Ltda. (Indaiatuba, SP, Brazil).

### 3.2. Preparation of Nanocapsules Containing Repellents

The nanocapsules were developed according to the interfacial deposition of pre-formed polymer method, first described by Fessi et al. [51] and labeled Nano_Control (without bioactive compound), Nano_GER1%, Nano_GER3%, Nano_ICAR1%, and Nano_ICAR3%. Briefly, two formulations were developed in which the organic phase was composed of the polymer poly-ε-caprolactone (PCL), acetone, caprylic acid, geraniol, and icaridin (100 mg and 300 mg). The aqueous phase was composed of polysorbate 80 (Tween^®^ 80) and deionized water. The organic phase was heated (~70 °C) to solubilize the polymer and then slowly poured into the aqueous phase. The solution was magnetically stirred and transferred to a rotary evaporator. The evaporated solvent was replaced with deionized water to obtain a final volume of 10 mL.

### 3.3. Characterization of Nanocapsules

#### 3.3.1. Particle Size, Polydispersity Index (PDI), and Nanoparticle Tracking Analysis (NTA)

Dynamic light scattering (DLS) determines the size, size distribution, and polydispersity index (PDI) of nanocapsules. For these analyses, the samples were diluted with deionized water. The analyses were performed in triplicate, and the results are expressed as mean ± standard deviation. A ZetaSizer Nano ZS 90 analyzer (Malvern^®^) was used to determine the zeta potential values with 1:100 dilutions of the samples in deionized water. The samples were diluted in deionized water (1:1000 *v*/*v*) and inserted into a cuvette for analysis. The results are expressed in mV, with the mean and standard deviation of three determinations. Nanoparticle tracking analyses (NTA) were performed using NanoSight LM14 equipment (green laser, 532 nm), where the images of the nanocapsules were collected by an sCMOS camera using NanoSight software version 2.3 (Malvern Instruments, UK). Initially, the formulations were diluted in deionized water (dilution factor: 1 × 10^4^). Subsequently, 1 mL of each formulation was injected into the volumetric cell to start the analysis of the nanocapsules. For each sample, approximately 143 particles/frame were obtained, resulting in a total of 1951 frames. Each study included five measurements.

#### 3.3.2. Differential Scanning Calorimetry (DSC)

The samples Nano_Control (without bioactive), Nano_GER1%, Nano_GER3%, Nano_ICAR1%, and Nano_ICAR3% were subjected to thermal analysis under a nitrogen atmosphere in the temperature range of 25 to 300 °C, a gas flow of 50 mL/min, and a heating rate of 10 °C/min using a DSC-Q20-TA Instruments device. For DSC, the formulations were analyzed in the range of 0 to 300 °C under a nitrogen atmosphere, with a gas flow of 50 mL/min and a heating rate of 10 °C/min. Aluminum sample holders and sample masses ranging from 2.0 to 4.0 mg were used.

#### 3.3.3. Fourier Transform Infrared Spectroscopy (FTIR)

Chemical characterization of the components and nanocapsules was performed by Fourier transform infrared spectroscopy (Shimadzu, IRAffinity, Kyoto, Japan) operating in the range 4000–400 cm^−1^, with a resolution of 4 cm^−1^, and 160 scans were accumulated with a resolution of 4 cm^−1^. The samples Nano_Control, Nano_GER1%, Nano_GER3%, Nano_ICAR1%, and Nano_ICAR3% were analyzed using the KBr method. Approximately 1 mg of the sample was mixed with 200 mg of potassium bromide (KBr) in an agate crucible until a fine and homogeneous powder was obtained. Subsequently, the powder was dried in an oven (~70 °C) for 20 min to remove any moisture. Then, the sample was placed on a support, subjected to a hydraulic press to form pellets, and sent for analysis.

#### 3.3.4. Encapsulation Efficiency

The encapsulation efficiency was determined by gas chromatography using an indirect method and expressed by the ratio between the difference between the concentration of a marker present in the oil (*CT*) and the free concentration of this marker in the supernatant (*CL*) divided by *EE* (Equation (1)).
𝐸𝐸 (%) = (𝐶𝑇 − 𝐶𝐿)/𝐶𝑇 × 100(1)

The detection and quantification limits of geraniol and icaridin (Table 2) were obtained through linear regression of the analytical curves. These results indicated that the data obtained were considered satisfactory because the values followed those recommended by Resolution of the Collegiate Board (RDC) 116/2007 by the Brazilian Health Regulatory Agency (ANVISA) which establishes criteria for the validation of analytical methods.

#### 3.3.5. Atomic Force Electron Microscopy (AFM)

The samples Nano_GER3% and Nano_ICAR3% were analyzed by atomic force microscopy using the Nanosurf EasyScan 2 AFM equipment to visualize the morphology and size distribution of the nanomaterials. Initially, 1 μL of the nanocapsule suspension was deposited on a silicon surface, and the samples were then dried in a desiccator for 48 h. For the analysis, the equipment was operated in non-contact mode with a TapAl-G cantilever (BudgetSensors, Bulgaria) and a peak voltage of 90 Hz. The scan speed was proportional to the scan area and scan frequency (0.6 Hz). The acquired images were analyzed using Gwyddion software.

### 3.4. In Vitro Release Assay

The release assay was performed according to the methodology described by Sotelo-Boyás et al. [52] with some modifications. For this test, the formulations Nano_GER3% and Nano_ICA3% were selected in a 1:1 proportion to verify the release profile of the nanostructured system and the application of mathematical models for evaluation of the reagent release mechanism. The emulsions containing geraniol and icaridin were developed by the low-energy emulsification method, which consisted of preparing the (organic) phases containing geraniol or icaridin (3% *v*/*v*) and the aqueous phase composed of distilled water (40 mL) and Tween 80 surfactant (60 mL). After solubilization, the aqueous phase was poured over the organic phase and kept under magnetic stirring for 30 min. A diffusion apparatus was used to carry out in vitro release study of the optimized formulations. The apparatus consisted of two compartments (donor and recipient) separated by a dialysis membrane (1 kDa exclusion pore size, Spectrapore). In the donor compartment, 1 mL of the formulations containing the nanocapsules of geraniol and icaridin was added and then immersed in the recipient compartment that contained a Tween 20^®^ solution 5% (*v*/*v*) and water in order to receive the actives that permeated through the membrane. The system was kept under constant magnetic agitation. Aliquots (1 mL) were collected from the recipient compartment periodically and stored in vials. The system was kept in a closed compartment to avoid possible losses by evaporation. The test was performed at a temperature of 32 °C, mimicking the skin temperature, the desired location for application. Analyses were performed in triplicate and quantification was by HPLC. After the quantification of the samples by HPLC, they were applied to determine the concentration later and later after the cumulative release (Equation (2)).
(2)$$CR=\frac{Conc.Final}{Conc.Initial}\times 100$$

### 3.5. Cytotoxicity and Genotoxicity Assay of Nanocapsules

#### 3.5.1. Mitochondrial Activity Assay—MTT (3-(4,5-Dimethylthiazolyl-2)-2,5-diphenyltetrazolium bromide)

Cytotoxicity analysis by the MTT method used the cell lines 3T3 (embryonic Swiss albino) and V79 (normal lung Chinese hamster). Initially, cells were plated at a concentration of 1 × 10^5^ cells/well in 96-well plates, which were incubated at 37 °C with 5% CO_2_. After 24 h, cells were exposed to Nano_Control and Nano_GER/ICAR3% formulations at different concentration ranges. The exposure lasted for 24 h, and incubation was at 37 °C and 5% CO_2_, decreasing the stock solution concentrations. After the treatment period, the nanocapsules were removed from the cultures, and the cells were washed with PBS, then 100 μL of MTT (3-(4,5-dimethylthiazolyl-2)-2,5-diphenyltetrazolium bromide) solution were added to each well at a concentration of 0.5 mg/mL and incubated for 3 h at 37 °C and 5% CO_2_. The MTT solution was then removed, and the cells were fixed by adding 100 µL of DMSO per well. Cell viability analysis was performed using a microplate reader at 540 nm.

#### 3.5.2. Genotoxicity Evaluation

The genotoxicity of the nanocapsules was analyzed using a comet assay, according to the methodology adapted from Singh et al. [53] and Collins [54]. For this analysis, cell lines 3T3 (embryonic Swiss albino) and V79 (normal lung Chinese hamster) were exposed to nanocapsules containing geraniol and icaridin (1:1) at concentrations of 0.005, 0.01, and 0.02 mg/mL. The cells were then incubated for 1 h. After exposure, the cells were homogenized in 0.8% low melting agarose and spread on slides prepared previously with 1.5% agarose.

After mounting, the slides were immersed in a lysis solution for 1 h, followed by neutralization. Subsequently, the slides were stored in electrophoresis buffer at 4 °C for 20 min, followed by running for 20 min at 1.6 V cm^−1^. At the end of the electrophoresis, the slides were dried, fixed, and stained with a silver solution. Analyses were performed under an optical microscope (40×), considering approximately 100 cells per slide, following the visual score criterion proposed by Collins et al. [31]. The damage index (ID) of each treatment was calculated by dividing the score of each slide by the number of cells analyzed [54,55,56].

### 3.6. Statistical Analysis

The experiments were conducted in triplicate, and the results obtained are presented as the mean and standard deviation using Origin 8.0 software. Analysis of variance (one-way ANOVA) was performed to compare the means between the formulations and the control (Nano_Control), and the Tukey test was also applied (*p* < 0.05). For cell viability analysis, GradPad Prism 8.0.1 software (Graph Pad Software Inc., San Diego, CA, USA) was used.

## 4. Conclusions

Although there are several products with repellent activity toward *Aedes aegypti* currently on the market, these contain mainly synthetic substances, and their repetitive application and indiscriminate use can cause several adverse reactions, especially in children and pregnant women. The nanoprecipitation technique with solvent evaporation proved to be technically viable in terms of performance in the development of nanocapsules containing geraniol and icaridin for topical application, with an average size between 260 and 314 nm polydispersity (<0.2), and physicochemical stability over 90 days, giving the system controlled release following Hixson–Crowell mathematical models and first-order kinetics. Evaluation of cell viability genotoxicity showed that a mixture of nanocapsules containing geraniol and icaridin at low concentrations showed compatibility and an absence of damage to genetic material in cell lines 3T3 and V79 at low concentrations, providing a good possibility for alternative products for repelling *Aedes aegypti*.

## Figures and Tables

**Figure 1 ijms-23-08317-f001:**
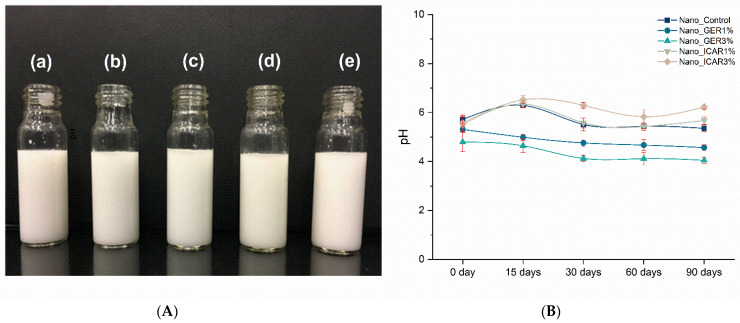
(**A**) Macroscopic view of the Nano_Control polymeric nanocapsule formulations: (a) Nano_Control; (b) Nano_GER1%—Nanocapsules containing 1% of geraniol; (c) Nano_GER3%—Nanocapsules containing 3% of geraniol; (d) Nano_ICAR1%—Nanocapsules containing 1% icaridin and (e) Nano_ICAR3%—Nanocapsules containing 3% icaridin. (**B**) pH profile of PCL polymeric nanocapsules containing geraniol and icaridin (0, 15, 30, 60 days, and 90 days). Measurements were performed in triplicate (*n* = 3).

**Figure 2 ijms-23-08317-f002:**
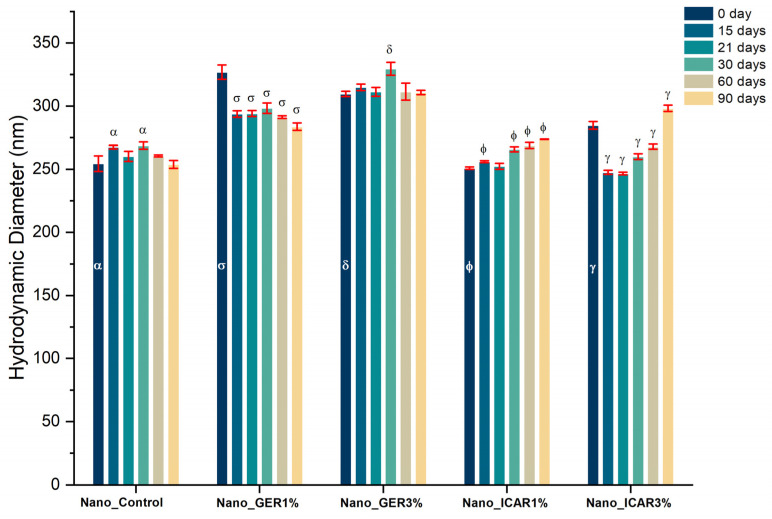
Hydrodynamic diameter (size) of polymeric nanocapsules containing geraniol and icaridin as a function of time (90 days). Measurements were performed in triplicate (*n* = 3); these values represent the mean of the three determinations. Considered significance of *p* < 0.05 (one-way ANOVA—Tukey) for the analysis of variance of times (15, 21, 30, 60, and 90 days) in relation to time 0. Equal symbols (α, σ, δ, Φ and γ) represent significant variation.

**Figure 3 ijms-23-08317-f003:**
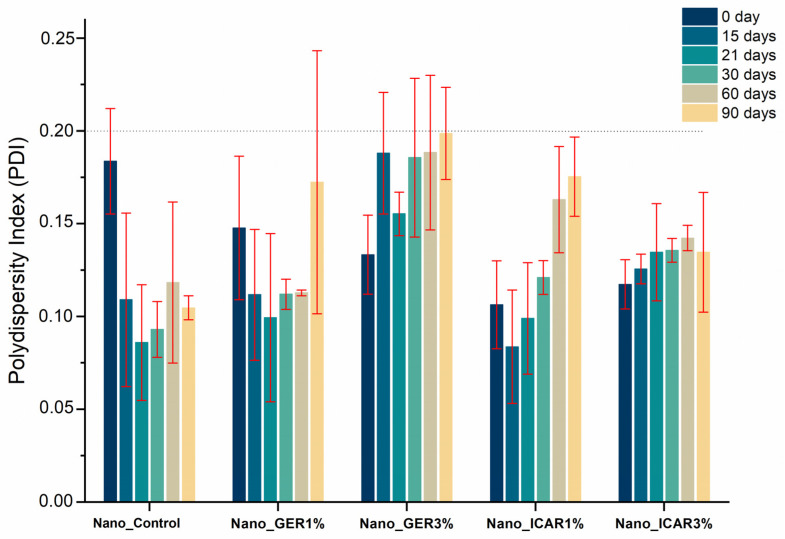
Results of polydispersity index (PDI) determinations of polymeric nanocapsules containing geraniol and icaridin as a function of time from 0 to 90 days. Measurements were performed in triplicate (*n* = 3).

**Figure 4 ijms-23-08317-f004:**
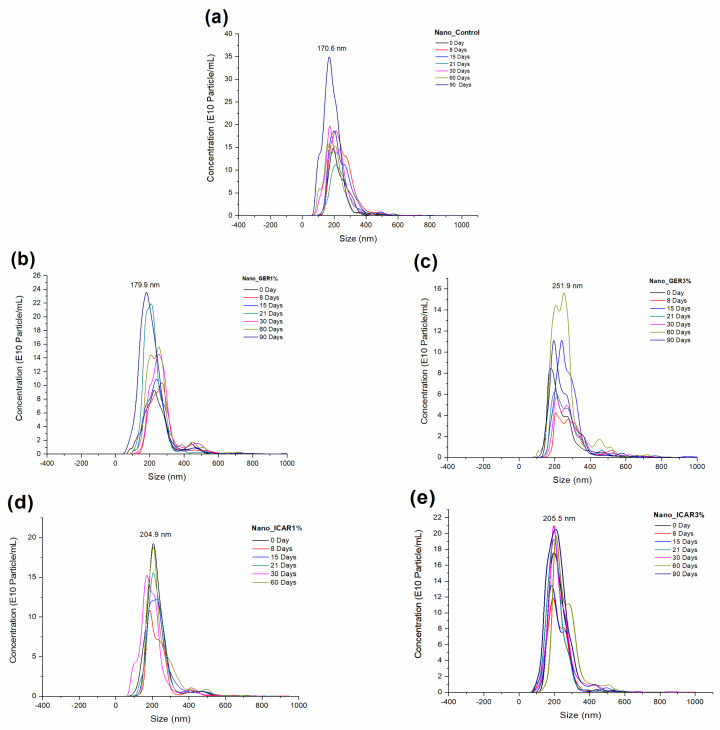
Particle size distribution as a function of concentration and time (0, 8, 15, 21, 30, 60, and 90 days) of the formulations (**a**) Nano_Control, (**b**) Nano_GER1%, (**c**) Nano_GER3%, (**d**) Nano_ICAR1%, and (**e**) Nano_ICAR3% using Nanoparticle Tracking Analysis (NTA).

**Figure 5 ijms-23-08317-f005:**
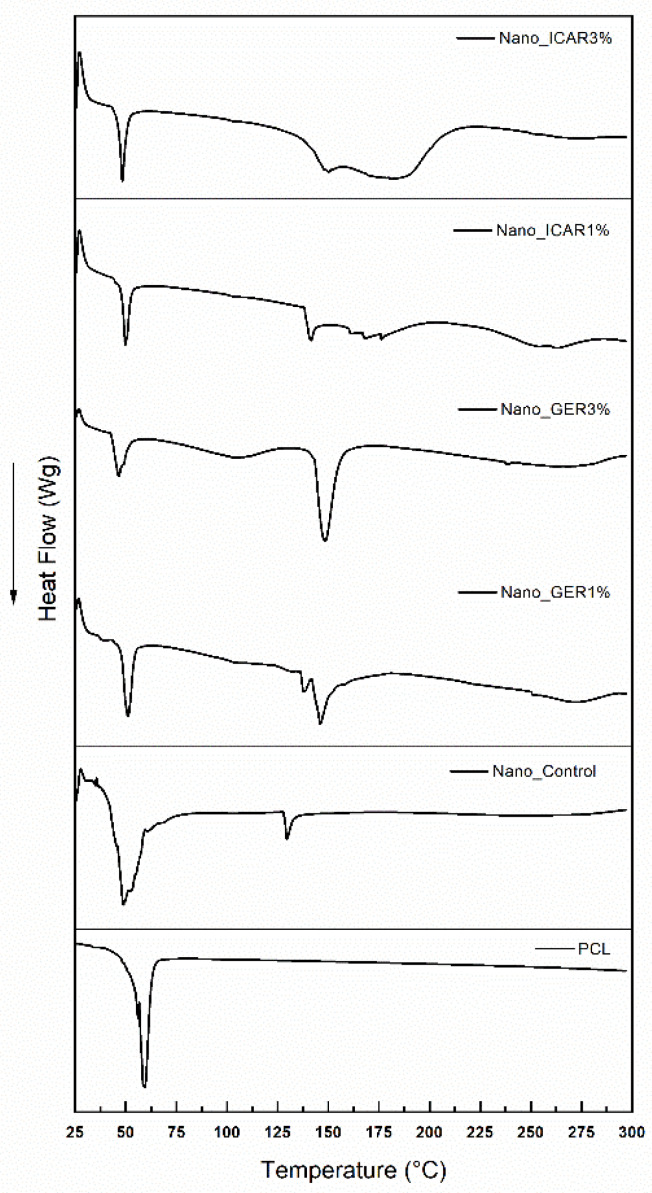
Differential exploratory calorimetry thermograms for: Nano_Control, Nano_GER1%, Nano_GER3%, Nano_ICAR1%, Nano_ICAR3%, and PCL.

**Figure 6 ijms-23-08317-f006:**
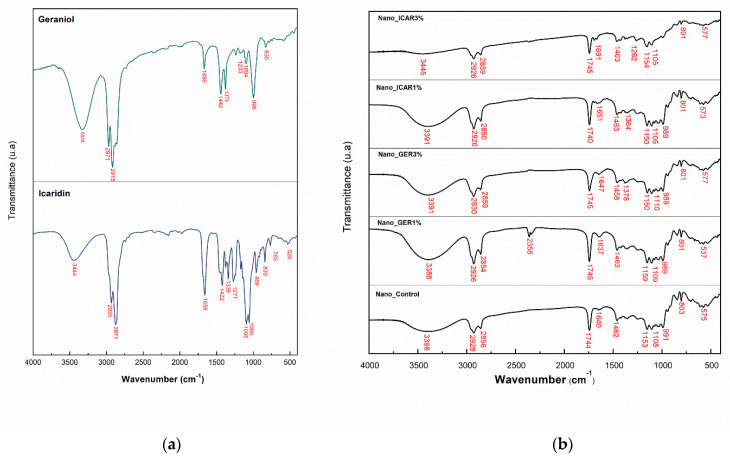
(**a**) Infrared spectra for the bioactives geraniol and icaridin and (**b**) infrared spectra for the formulations Nano_Control, Nano _GER1%, Nano_GER3%; Nano_ICAR1%, and Nano _ICAR3%. The spectra were obtained through Fourier transform infrared spectroscopy (FTIR) using KBr, in the frequency range from 4000 to 400 cm^−1^.

**Figure 7 ijms-23-08317-f007:**
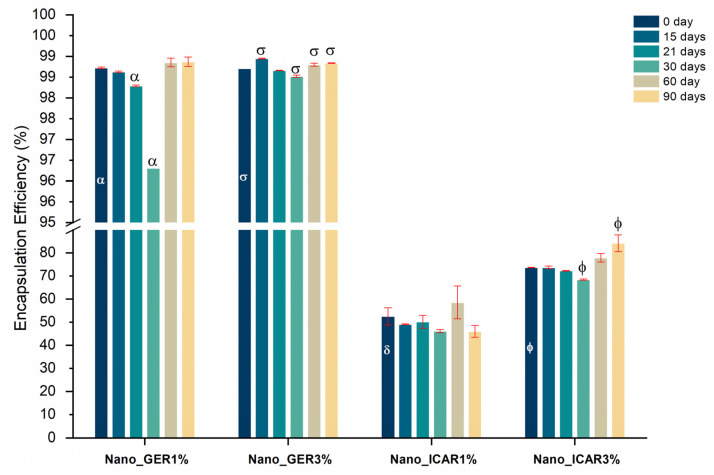
Encapsulation efficiency (%) of polymeric nanocapsules as a function of time (0, 15, 21, 30, 60, and 90 days). Measurements were performed in triplicate (*n* = 3); these values represent the mean of the three determinations. Considered significance of *p* < 0.05 (one-way ANOVA—Tukey) for the analysis of variance of times (15, 21, 30, 60, and 90 days) in relation to time 0. Equal symbols (α, σ, δ and Φ) represent significant variation.

**Figure 8 ijms-23-08317-f008:**
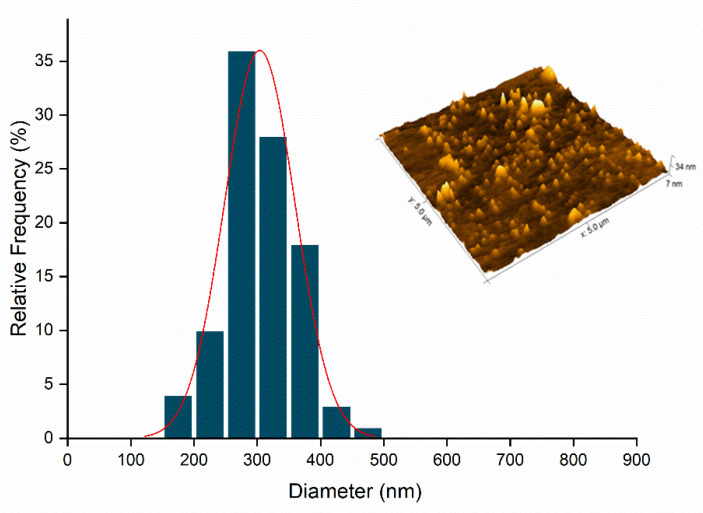
Representation of the size distribution graph (nm) in relation to relative frequency (%) and topography referring to the Nano_GER3% formulation.

**Figure 9 ijms-23-08317-f009:**
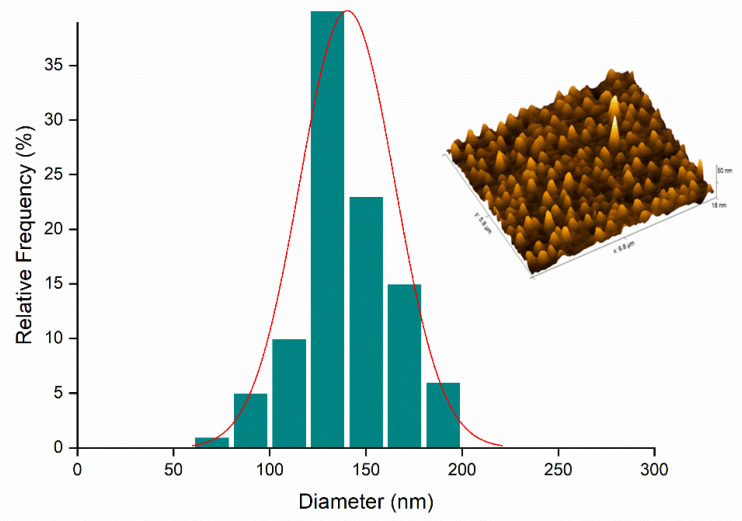
Representation of the size distribution graph (nm) in relation to relative frequency (%) and topography referring to the Nano_ICAR3% formulation.

**Figure 10 ijms-23-08317-f010:**
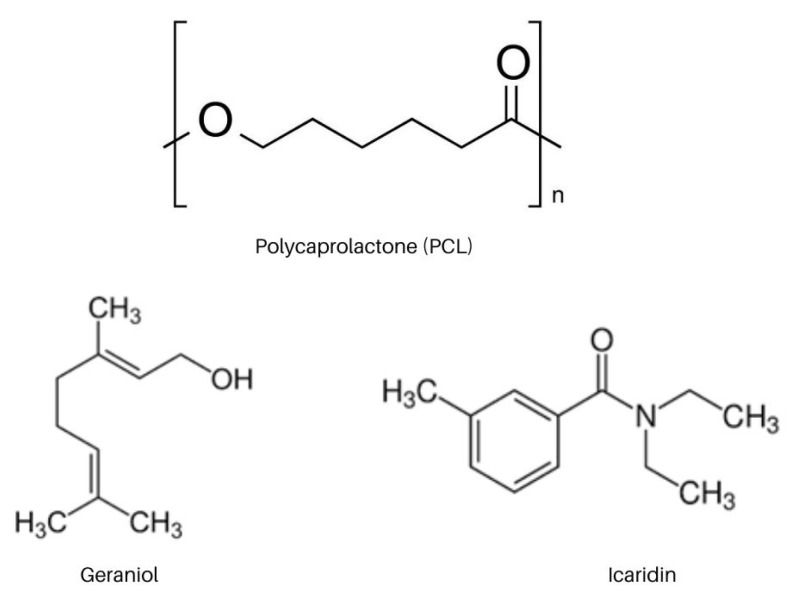
Chemical structures of Poly-ε-caprolactone (PCL), geraniol and icaridin.

**Figure 11 ijms-23-08317-f011:**
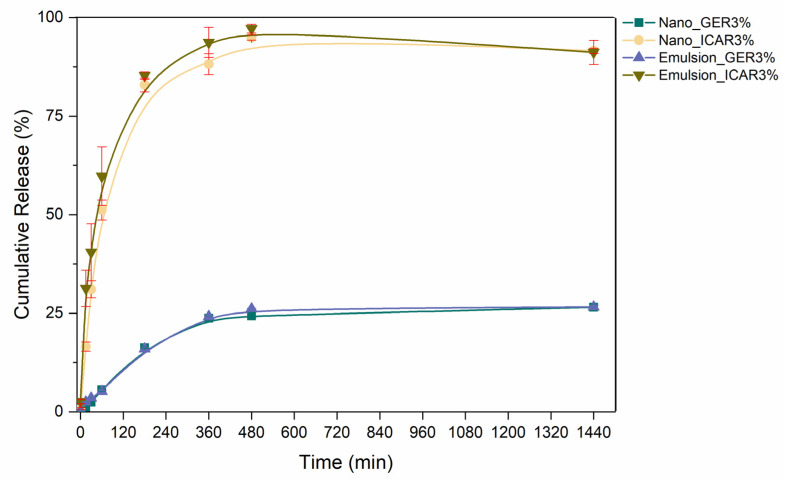
The graph shows cumulative release curves (%) of Nano_GER3% nanocapsules: Nano_ICAR3%, Emulsion_GER3%, and Emulsion_ICAR3% were carried out at a temperature of 32 °C. The analyses were performed in triplicate (*n* = 3) and quantification by high-performance liquid chromatography (HPLC).

**Figure 12 ijms-23-08317-f012:**
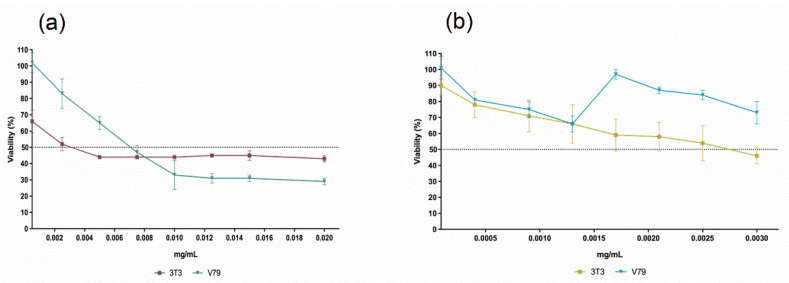
Cytotoxicity evaluation of formulations (**a**) Nano_GER3%/ICAR3% and (**b**) Nano_Control on 3T3 and V79 cell lines.

**Figure 13 ijms-23-08317-f013:**
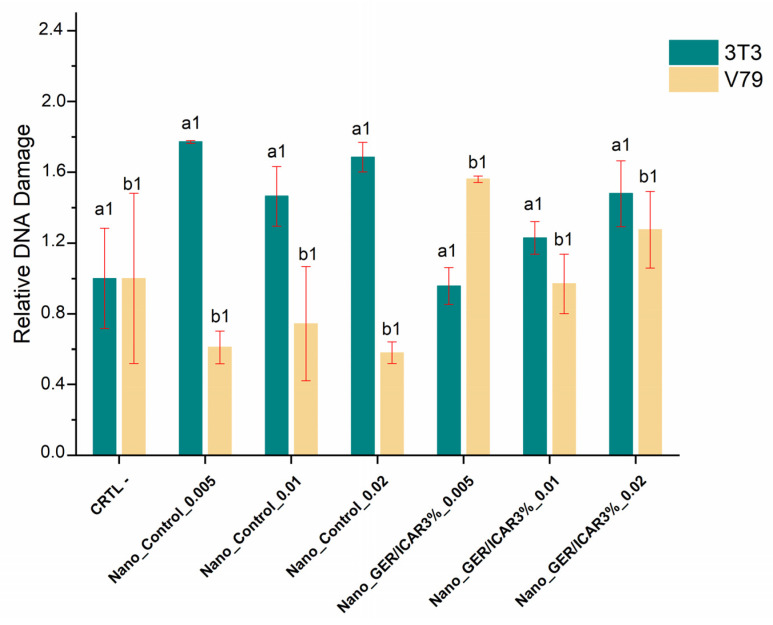
DNA damage assessment by the comet assay in cell lines: 3T3 and V79 exposed to concentrations of 0.005; 0.01, and 0.02 mg/mL of the formulations Nano_GER/ICAR3% and Nano_S/Active. Equal numbers (a1, b1) are not statistically significant (*p* > 0.05).

**Table 1 ijms-23-08317-t001:** Correlation coefficient (R^2^) for the different mathematical models used for the release kinetics of nanocapsules containing the active substances geraniol and icaridin.

	Mathematical Models								
Samples	Higuchi		Korsmeyer–Peppas			Hixson–Crowell		First Order	
	R^2^	k	R^2^	k	n	R^2^	k	R^2^	k
**Nano_ICAR3%**	0.891	8.730	0.941	16.259	0.269	0.964	0.0051	0.930	0.0187
**Nano_GER3%**	0.907	4.556	0.822	1.4308	0.436	0.999	0.0036	0.998	0.0114

R^2^ = Correlation coefficient; k = release constant.

**Table 2 ijms-23-08317-t002:** Limit of detection and limit of quantification of the analytical methods for geraniol and icaridin.

	Geraniol	Icaridin
Limit of Detection (μg/mL)	0.39 ± 0.02	2.35 ± 0.86
Limit of Quantification (μg/mL)	1.30 ± 0.32	7.83 ± 1.65

## Data Availability

All data in this study will be available from the corresponding author upon reasonable request.

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
