# Peer review of "Polymeric Nanorepellent Systems Containing Geraniol and Icaridin Aimed at Repelling Aedes aegypti"

_ijms, 2022, doi:10.3390/ijms23158317_

Round 1

Reviewer 1 Report

Manuscript "Polymeric nanorepellent systems containing geraniol and icaridin aimed at repelling Aedes Aegypti" is well planned and described, and requires no major corrections except for some minor improvements that are listed below:

1. in introduction there is no explanaition why geraniol and icaridin were selecetd for study

2. the authors did not write how they determined concentration of these compunds

3. in the first lines of R&D the authors should introduce the type of formulation and indicated the goal of study

4. in Fig 2 there is not given PDI parameter (... Hydrodynamic diameter (size) and Polydispersity Index (PDI) of Polymeric...) ????

5. it would be necessary to introduce an axis slice for Fig 2 and 8 to increase the visibility and improve possibility of intrepretation of the results and their errors

6. What means "recommended by RDC 116/2007 " this should be explained

7. In release study why "The receiving box was formed using a 5% Tween 80 solution 466 kept under agitation."? what was exactly the releasing buffer?, why Tween 80 was used? what was its volume? ...?. This procedure should be decribed more precisely. The formula for cumulative release % calculations should be given.

8. why comet assay was not carried out for emmulsions or geraniol/icaridin to prove the advantage of the applied nanoformulations. How the emulsions were prepared?

9. chemical formulas of geraniol and icaridin should be given and possible interactions with polymeric matrice in nanoformulation should be discussed to give impression of possible release mechanism. Discussion on this taking into account fitted kinetic models should be introduced.

After taking these comments into account, I recommend this article for publication.

Author Response

Dear reviewer,

Reviewer 2 Report

The authors reported polymeric nanorepellent systems containing geraniol and icaridin with good physicochemical characteristics and good stability, and potential repellent activity against Aedes aegypti. However, there are certain concerns about figures and experiments. Therefore, I will reconsider accepting this work once the following questions are well addressed.

1. Please improve the figure format in this manuscript. For example, Figure 2 only shows hydrodynamic diameter data, but the caption mentioned Polydispersity Index as well. Figure 6 is missing.

2. Figure 10 and ‘2.6 In vitro release’ contain Emulsion_GER3% and Emulsion_ ICAR3% but no information regarding their preparation or representation is provided.

3. Some experimental design issues existed. For example, nanoparticles were not washed with water following nanoprecipitation based on the descriptions from Experimental Method. If the ‘In vitro release assay’ was conducted with a solution containing both nanoparticles and free drug, the initial burst release was caused by the unencapsulated drugs rather than the nanoparticles. This consequence can explain why Nano_GER3% (or ICAR3%) and Emulsion_ GER3% (or ICAR3%) showed similar release profiles in Figure 10. In addition, the encapsulation efficiency is measured once the nanoparticles are formed, thus, please explain how and why the encapsulation efficiency of Polymeric Nanocapsules over longer durations (15, 21, 30, 60, and 90 days) was measured in Figure 8.

4. The author used Nano_GER3%/ICA3% formulation for the evaluation of cytotoxicity and genotoxicity. Please provide a justification for selecting this formulation.

5. Figure 11 demonstrates cytotoxicity at different concentration ranges. Please specify if the concentration refers to drug concentration or nanoparticle concentration.

6. The pH of the polymeric nanocapsule suspensions experiments in Figure 1b should be performed in triplicate.

Author Response

Dear reviewer,

Reviewer 3 Report

The manuscript 'Polymeric nanorepellent systems containing geraniol and icar-2 idin aimed at repelling Aedes Aegypti' describes an alternative approach to produce polymeric nanorepellent systems.  

The authors need to improve on the following aspects to enhance the significance of the content.

1. Introducing the components of the formulations while introducing the approach in the introduction can help the reader understand the spirit of the research better. 

2. The results & discussion part starts abruptly. Reader would definitely want to now how and what type of particles are formed especially when the methods section is at the end of the manuscript.

3. While discussing the particle diameter and its trends, the authors should not rely on the DLS analysis only. Samples for DLS are typically diluted to a large extent and cannot truly represent all the particles. It is advisable to conduct nd report SEM analysis of the formulations while discussing the size and variation between formulations and after storage.

4. Page 7, Line 184: what do the authors mean by 'It is 184 not feasible to compare the results of these methods'? Please extend this sentence further.

 5. Page 8: A whole paragraph is assigned to the discussion of related literature and not the results of this study. It is advisable here that the discussion of other's results should be reduced to a couple of sentences instead.

6. Page 9, Like 219: What do the authors refer to 'The spectra (Figure 6) obtained from the bioactive geraniol and icaridin'. 

7. Page 13, Line 296: What do the authors mean by free active? Did they take a liquid formulation to study the release? If that is the case, they are suggested to study the release of actives form the solid particles only.

8. Page 17, Line 405. What si the dilution factor?

9. The text and labelling in some figures such as Figure 4 and 11are hard to understand. Please replot better quality figures. Also what is the significance of Figure 1 a? 

10. There are some careless formatting mistakes, e.g., Why is there a sentence 'The introduction should briefly place the study in a broad context and highlight why it is important' , on Page 1, line 31? or why a ',' is used instead of a '.' sign at most of the places, e.g., line 38, 40, Table 1.

11. Page 9, 1st paragraph: Why -OH is not written properly?

Author Response

Dear reviewer,

Round 2

Reviewer 2 Report

The authors have addressed the questions well and the manuscript can be accepted in its present form.